# Detection of Flexible Pavement Surface Cracks in Coastal Regions Using Deep Learning and 2D/3D Images

**DOI:** 10.3390/s25041145

**Published:** 2025-02-13

**Authors:** Carlos Sanchez, Feng Wang, Yongsheng Bai, Haitao Gong

**Affiliations:** Ingram School of Engineering, Texas State University, San Marcos, TX 78666, USA; cds242@txstate.edu (C.S.); f_w34@txstate.edu (F.W.); h_g153@txstate.edu (H.G.)

**Keywords:** pavement surface distresses, cracking, deep learning, YOLOv5, 2D/3D images

## Abstract

Pavement surface distresses are analyzed by transportation agencies to determine section performance across their pavement networks. To efficiently collect and evaluate thousands of lane-miles, automated processes utilizing image-capturing techniques and detection algorithms are applied to perform these tasks. However, the precision of this novel technology often leads to inaccuracies that must be verified by pavement engineers. Developments in artificial intelligence and machine learning (AI/ML) can aid in the progress of more robust and precise detection algorithms. Deep learning models are efficient for visual distress identification of pavement. With the use of 2D/3D pavement images, surface distress analysis can help train models to efficiently detect and classify surface distresses that may be caused by traffic loading, weather, aging, and other environmental factors. The formation of these distresses is developing at a higher rate in coastal regions, where extreme weather phenomena are more frequent and intensive. This study aims to develop a YOLOv5 model with 2D/3D images collected in the states of Louisiana, Mississippi, and Texas in the U.S. to establish a library of data on pavement sections near the Gulf of Mexico. Images with a resolution of 4096 × 2048 are annotated by utilizing bounding boxes based on a class list of nine distress and non-distress objects. Along with emphasis on efforts to detect cracks in the presence of background noise on asphalt pavements, six scenarios for augmentation were made to evaluate the model’s performance based on flip probability in the horizontal and vertical directions. The YOLOv5 models are able to detect defined distresses consistently, with the highest mAP50 scores ranging from 0.437 to 0.462 throughout the training scenarios.

## 1. Introduction

The condition scores of a pavement section are evaluated by the amount and severity of surface distress it contains. Transportation agencies use the condition scores to determine when a roadway section should receive maintenance and rehabilitation (M&R), strategically extending the pavement’s lifespan and improving its performance. Condition data collection is crucial for this process, as jurisdictions can contain hundreds of thousands of lane-miles. There are three methods to obtain pavement surface data: manual, semi-automated, and automated data collection.

Historically, manual data collection occurred at a sampled portion of the pavement section, where raters directly measured the length and severity of the surface distresses. This process posed a risk to the pavement raters due to exposure to traffic and harsh weather, along with inaccuracies due to human error. Semi-automated data collection could collect surface data along the entire length of the pavement section using image-capturing equipment attached to a survey vehicle. Once collected, the data would be viewed by pavement engineers and measured for distress. With automated data collection and evaluation, the measurement and severity of the pavement surface would be determined using algorithms that would interpret the visual data. However, this novel technology often led to inaccuracies, and the condition evaluations would have to be verified by the pavement engineers. To evaluate the effectiveness of each data collection and evaluation method, researchers compared the accuracy and applicability of semi-automated and fully automated systems. Cheng and Miyojim successfully developed a skeleton analysis algorithm coupled with image processing techniques that could detect and classify distresses along with their severity [1]. This novel approach to automated evaluation was further analyzed by Tighe et al. by using laser sensor and digital image capturing equipment, finding the method to be equivalent to manual data collection but should be supplemental for condition evaluation [2]. An issue with automated data collection is the data quality along with the technology used in the collection and evaluation of the pavement surface. Luo et al. used an online questionnaire survey to investigate the techniques used in pavement surface data collection conducted by several state and local highway agencies. They had found there was no uniform protocol for automated or semi-automated data collection, data quality provided a challenge to the agencies, and novel technologies such as artificial intelligence (AI) would have to be further optimized. The improvement in AI would have to occur prior to being considered for use on a pavement network scale [3].

Since pavement surface data are evaluated using visual data, an issue with image quality in automated evaluation is in the use of 2D images. Non-distress objects such as liquid stains and debris often cause confusion for detection algorithms to incorrectly classify them as distresses. Subtle distinctions of features between objects provide difficulty for AI models, but they can be circumvented with proper techniques. Qi et al. utilized a novel visual blurriness network to properly detect glass surfaces from image data based on its intrinsic property to slightly distort light [4]. Along with this, visible light has its limitations when visualizing these subtle features. Other frequencies of visible data can provide much-needed context for AI models, as Tang et al. studied. Their research incorporated modalities that processed visual RBG and thermal data separately before integrating both for feature extraction, successfully improving the detection performance of their model [5]. With 3D image capture, surface depth is taken into consideration and provides further context that can extract more features from the pavement surface. Ouyang and Xu were able to detect and capture surface cracks with a 3D camera and laser sensors, yielding a 2% difference in crack length on a single section to prove the repeatability of this technology [6]. Huang et al. combined features of both 2D and 3D images to determine the accuracy of their algorithms with both image types along with the fusion of the two. They successfully improved the accuracy of the algorithm along with minimizing the uncertainty in detection using the Dempster–Shafer Theory [7].

Developments in the field of artificial intelligence and machine learning (AI/ML) provide aid in the efforts to improve automated pavement surface data collection and evaluation. Zhang et al. used ConvNet with 500 images in a 3264 × 2448-pixel resolution for distress detection, yielding a precision score of 0.87 with recall and F1 scores of 0.975 and 0.8971, respectively [8]. Zhang A. et al. developed a CrackNet model to compare its performance with a Pixel support vector matrix (SVM) and 3D shadow modeling, with scores ranging from 0.8793 to 0.9013. While successful, the CrackNet model found issues with detecting hairline cracks, which had a width of a few pixels [9].

One type of AI/ML model suited for this task is convolutional neural networks (CNNs), which can interpret visual data through convolutions with a kernel that will comb through the image pixels to extract feature maps. Tong et al. combined CNNs and a ground penetrating radar (GPR) to detect and measure concealed cracks along with subsurface voids, surface distresses, and subgrade settlement. Their model performed well in detection and classification but was unable to use 3D reconstruction to measure the damage [10]. Region-based convolutional neural networks (R-CNN) are CNNs that separate region proposals and classification to fulfil object detection tasks with improved accuracy. Song and Wang developed and evaluated 20 versions of Faster R-CNN based on their region proposal network (RPN) anchors. Faster R-CNN sends the feature maps created from the convolution layers to the RPN, where region proposals are made so that the model focuses on those regions of interest (RoI) instead of the entire image, improving accuracy. Their most successful model held a precision score of 0.915, with a pixel location error of 6.271 pixels [11].

The You Only Look Once (YOLO) models, developed by Ultralytics which is located in Frederick, MD, USA, are a line of deep learning detection models that balance speed and accuracy [12]. Du et al. compared a YOLOv3 model with Faster R-CNN and found that the two had similar detection accuracies but struggled with potholes and fatigue cracking [13]. Ren et al. used several attention modules, such as the convolutional block attention module (CBAM) and coordinate attention (CoordAtt), to modify a standard YOLOv5 model and improve its accuracy [14]. Gong et al. compared YOLOv5 with Faster R-CNN and CenterNet to investigate influential factors that affect a model’s performance on a dataset. After the F1 scores were raised from 0.7 to 0.84 for YOLOv5 and Faster R-CNN, it was determined that a sufficient dataset size, consistent annotation work, and distress detection was a unique version of general object detection that required more effort to achieve adequate scores [15]. Quan et al. developed a Central Feature Pyramid (CFP) network to focus on features in corner regions of input images to improve predictions on the MS COCO 2017 dataset. Modifications to YOLOv5 and YOLOX were made using this CFP network, with an objective for utilizing deep features to regulate shallow, central features. This implementation of the CFP network provides enhanced predictions in the models [16]. Wang et al. collected 2000 pavement image sheets to train a YOLOv8 model along with a modified YOLOv8 with a CBAM neck and DarkNet53 backbone. The modified YOLOv8 held scores ranging from 0.982 to 0.995 [17].

The use of CNNs for pavement distress detection and evaluation has shown to have promising results. With large datasets and proper training techniques, deep learning models can be developed with adequate performance to resolve the issues with inaccuracies found in automated pavement surface distress detection and evaluation. This is essential in areas such as coastal regions, where rapid development and exposure to harsh environmental conditions are common. Pavement sections in coastal regions are subject to saltwater corrosion, heavy rainfall, extreme weather phenomena, and flooding at higher rates than in sections found inland or away from large bodies of water [18,19,20,21]. These factors that facilitate pavement deterioration and development of surface distresses provide a cause for concern in M&R application. Roadways in coastal areas serve vital functions in terms of evacuation, supply chain, and commercial routes and can benefit from the implementation of this novel technology so that adequate and appropriate M&R is distributed.

This research aims to develop a robust deep learning detection model using a library of labeled 2D/3D pavement images. The data quality, scope, and size of the dataset will introduce a unique approach to the model’s development. This paper’s Section 2 will discuss how the pavement surface data were collected and annotated, the selected model, and how the model’s performance was evaluated. The training scenarios for the model’s development are introduced in Section 3. The performance of the model is presented in Section 4, followed by the findings of this study in Section 5.

## 2. Methodology

### 2.1. The 2D/3D Pavement Data Collection

A robust dataset consisting of high-quality images with diverse amounts of distresses is needed to properly develop and train deep learning models. A custom dataset is formed by collecting miles of images of the pavement surface using the pavement survey vehicle, as shown in Figure 1. Attached to the rear roof of the vehicle is a HyMIT laser sensor from HYMIT LLC in Austin, TX, USA, which captures visual data of the road surface at a height of approximately 2.1 m above ground level with 2D images and 3D laser scans as the vehicle was driven. A 2D image is a simple photograph of the road surface, while 3D images are constructed using laser scans. The images are captured using a strong line laser, covering a width of 4 meters (m) in the transverse direction and close to 15 m in the longitudinal direction [6]. The sensor is coupled with an AMES Pro GPS-DMI from Ames Engineering in Ames, IA, USA so that the line laser is emitted at frequency intervals of 7 mm in the transverse direction and 1.04 millimeters (mm) in the longitudinal direction.

Visual data captured are extracted and split into two separate image files containing either the 2D or 3D image of the pavement surface. Both image formats are captured due to the advantages and disadvantages both provide when evaluating surface conditions. The 2D images can visualize surface type and crack seals but struggle with thin or shallow distresses. The 3D images utilize depth to provide contrast between the surface and regions with lower or higher distances from the surface. The images are resized into grayscale images at a size of 4096 × 2048. The pixel intensity in the 3D images is determined by the distance between the laser scanner and individual pavement surface. This distance is also called the depth. Darker pixel regions on a 3D image mean larger distance (depth) but indicate a potential distress of the pavement, whereon the contrary, lighter pixel values represent shorter distances and appear in the locations of lane markers, crack seals, and debris. As the vehicle is driven along the road, the laser line scans the road surface multiple times per second at a set frequency to extract as much detail as possible. The laser’s frequency that are used to capture the images determine the details of the distresses, especially with hairline cracks. These thin cracks can appear as a few pixels wide, which are also a few millimeters wide in reality.

Locations for the data collection sites are displayed in Figure 2, where emphasis is placed on pavement sections with closer proximity to the coast of the Gulf of Mexico. The pavement sections from the states of Louisiana and Mississippi consist of 408 miles of mostly flexible pavement containing around 40,000 images. Image data from sections collected in Texas contain 30 miles of flexible pavement surface with 3392 images.

The ground-truths for the models are made with manual annotations via bounding boxes. A custom software program [15] is used to create these bounding boxes to locate and label the defects across the dataset. The interface of this in-house developed program is shown in Figure 3, where the 2D and 3D images are displayed next to each other in a manner that represents the longitudinal length of the road surface as the pavement survey vehicle captured data. The class list in Table 1 is used to label the distresses and common objects visible in the images, as shown in Figure 4. The distresses can appear as singular cracks either parallel or perpendicular to the road’s centerline. They can also appear as an area of sparsely or tightly connected cracks, such as block cracks or alligator cracks, respectively, or deep, dark regions on the surface in the dataset. Seven of the nine labels used are distresses, with joints and lane longitudinal cracks being commonly detected objects due to their shared geometry of transverse and longitudinal cracks, respectively. The 3D images are primarily used for the annotations due to their visualization of the distresses.

Currently, the dataset consists of 3855 images of flexible pavement covering a distance of 34.4 miles. This subset of the full dataset was selected based on image quality and acquiring a substantial amount of instances for each distress class. The distribution of the labels for distresses and non-distresses is shown in Figure 5. The most prevalent distresses are transverse and longitudinal cracks, as they contain close to 50% and 25% of the total objects within the dataset, respectively. YOLOv5 has extensive community support stemming from years of development along with its ease of use due to its maturity and hardware compatibility. In addition, our previous research study in a similar scenario [5] showed that YOLOv5 was the best-performing model for our dataset and the computational environment setting. It is in our plan to apply the YOLOv5 and other models on the image data of different resolution levels using different data augmentation techniques to decide the best strategy for our next study, especially with limited computational resources. The high number of these two distresses are due to both transverse and longitudinal cracks being the most common forms of distresses across multiple highway networks.

### 2.2. Model Selection

This study aims to use CNNs due to their capabilities with visual data and image pixel values. YOLOv5, a state-of-the-art detection model, was chosen to be trained and developed with the current dataset [22]. Although there are newer versions of YOLO, such as YOLOv8 and YOLOv11, the extensive literature and research of YOLOv5 for pavement distress detection made it a great candidate due to its real-time detection and classification capabilities. YOLOv5 can also receive modifications to its architecture later on to improve its performance and is not as novel as YOLOv11, which was released in September 2024. Developed by Ultralytics, the architecture of the standard YOLOv5 is displayed in Figure 6, where the cross-stage partial network (CSPNet) backbone and Path Aggregation Network (PANet) neck process the input image in a single pass, balancing speed and precision.

In the backbone, BottleNeckCSP is used to split the input image into two parts. One part of the input is sent through convolution, while the other part goes through convolution, bottleneck, and concatenation layers. This process, visualized in Figure 7, reduces the computational load by allocating most of the feature extraction to one part rather than the whole image. With the PANet neck, SPP aggregates the input information and produces a fixed length for the output. This process improves the speed of YOLOv5 without sacrificing accuracy. The head portion of YOLOv5 consists of three convolutional layers to determine coordinates for the predicted bounding boxes, scores, and object class. The COCO dataset contains 80 common object classes but does not contain labels or tasks for pavement distress detection. However, similar studies like those of Gong et al. [15] show that YOLO models can have effective transfer-learning between the COCO dataset and pavement distress detection. This study uses the same strategy. In addition, data augmentation techniques are introduced into this research to increase the number of training data, reduce the imbalance problem among the annotated data, and improve the robustness of our models. Activation functions such as the sigmoid linear unit (SiLU) are used to introduce nonlinearity during convolution, while sigmoid activation is introduced at the end of feature extraction for object class probability [22,23]. YOLOv5 is user-friendly and comes with multiple packages to implement training, validation, testing, data augmentation, and logging. It is compatible with the PyTorch machine learning library and is pretrained with the Common Objects in Context (COCO) database, allowing for smoother integration with most datasets for object detection and classification [24].

### 2.3. Performance Metrics

The key metrics to determine the efficiency of a deep learning model are precision, recall, and F1 scores. Equations (1) and (2) contain the formulas for each metric, taking three variables into account: true positives (TP), false positives (FP), and false negatives (FN) [25]. True positives are when the model’s predictions align with the ground-truth from the manual annotations. False positives are when an incorrect detection or classification by the model is made that was not established in the ground-truths of the dataset. False negatives occur when the model does not detect an object that is present in the manual annotations.
(1)$$Precision=\frac{TP}{TP+FP}$$
(2)$$Recall=\frac{TP}{TP+FN}$$

Precision evaluates the model’s ability to correctly detect and identify objects, while recall evaluates its capabilities in detecting all relevant objects within the dataset. Another metric that is often used by YOLOv5 is the mean average precision (mAP), shown in Equation (3) [14]. It combines the previous metrics by plotting precision against recall to determine the area under the curve as average precision.
(3)$$mAP=\frac{1}{n}\sum_{i=1}^{n}{AP}_{i}$$

With YOLOv5, the mAP scores of each class are made available after validation with intersection over union (IoU) thresholds of 50% and 50–95% overlap between predicted and ground-truth bounding boxes. The context given from the individual mAP values can provide insight into where YOLOv5 exceeds well and where it faces difficulties in distress detection. Along with this, the scores can reflect the strength of the dataset as overfitting and class imbalances may influence the model’s performance.

## 3. Experimental Setup

In this research, the YOLOv5 model is developed with the current dataset at a distribution of 8:1:1 for training, validation, and testing. It is also trained for at least 300 epochs, and the learning rates during training ranged from 0.01 to 1 × 10^−5^. The epochs determine how many passes of the dataset the model will see, while learning rates establish how often the model adjusts its weight for a given number of steps. Batch sizes are used by the model to process a set amount of data inputs before the parameters are updated. Augmentations can artificially expand a dataset by adjusting and slightly altering data entries via transformations, rotations, or other image editing techniques.

Six scenarios are established for the YOLOv5 model in order to evaluate the most effective data augmentation techniques that could benefit the model’s performance. The change in the model’s training parameters and augmentations are shown in Table 2. Scenario 1 has little to no deviations from the original training script provided by Ultralytics except for changing the batch size to 4. Scenario 2 raises the batch size to 32. Beginning with Scenario 3, the image size is not altered from 1024 × 1024 pixels; this scenario has a batch size of 16. After Scenario 4, the batch size is not changed from 8. In Scenario 4, all the augmentations in the hyperparameter script are activated, as seen in Table 3. These augmentations include hue, saturation, and brightness values; geometry via scaling; translation; angularity; and perspective change on the input images.

In addition to the activation of these augmentations, the flip probability in both directions is raised from 0 to 0.5. All augmentations except for vertical and horizontal flips were deactivated in Scenario 5. For Scenario 6, the flip probability is raised from 0.5 to 0.75 in both directions.

The restrictions on the image size and batch size in Scenarios 3 and 4, respectively, are due to hardware limitations. An NVIDIA Quadro P4000 GPU is used to facilitate the training of the proposed YOLOv5 in an efficient manner with the current dataset. It is the available memory that restricts the increase of image size during the training and the local storage space that keeps batch size from doubling beyond eight training samples. Despite these issues, the developed YOLOv5 could process the dataset images and perform consistently throughout most of the training scenarios.

## 4. Initial Results and Discussion

When given the opportunity to train without issues due to memory depletion, YOLOv5 was quite efficient in balancing speed and precision. An average training session with the image size at 1024 × 1024 took 14–18 h to complete. The inference speed for the predictions was 34.3 ms per image. This is due to YOLOv5’s ability to process the input images within a single pass. As mentioned previously, hardware limitations due to memory and storage space depletion resulted in smaller image sizes, most likely influencing the model’s detection results. In Table 4, the mAP50 values of each of the six scenarios are shown. This table includes scores for each class label along with the model’s general performance.

With the exception of Scenario 4, where all default augmentations in Table 3 are activated, the scores are consistent throughout the sessions. The implementation of the flip probability provides the most significant improvement in the model’s detection. In Scenarios 5 and 6, where the flip probability was 0.5 and 0.75, respectively, six of the nine classes had the highest mAP50 scores. At the probability of 0.75, block cracks have their highest mAP50 scores, at 0.177. Joints have the highest mAP50 scores at around 0.80, while block cracks have the lowest at around 0.12. This is because joints are wide enough but straighter and more linear to be captured by the laser scanner than other cracks, which makes their characteristics unique for the model to detect them. As for block cracks, there are fewer instances than other distresses due to the imbalance of the dataset on longitudinal cracks, transverse cracks, and the rest of the classes. Block cracks also lack key defining features from alligator cracks and failures, such as large areas of crack webs or deep spots. Meanwhile, alligator cracks and failures could be similar to each other on some annotated images. Failures, despite having unique characteristics in the form of potholes, have multiple appearances that vary based on the distress width. The limited number of failures in the current dataset presents a wide variety of appearances, such as features of slightly widened cracks, leading to their highest mAP50 scores being 0.562. These variations caused confusion for the proposed YOLOv5 models in effectively detecting the failures as deep and eroded surface regions. The detection of longitudinal and lane longitudinal cracks performed similarly to each other due to sharing similar features as opposed to joints, with scores of 0.436 and 0.476, respectively. Because their erratic patterns in each direction on the pavement surface make it difficult to detect properly, both sealed crack types are poorly detected with the trained YOLOv5 models, yielding scores of 0.473 for sealed transverse cracks and 0.325 for sealed longitudinal cracks. Some of the predictions are shown in Figure 8, displaying the detection capabilities of the models.

Besides the distinct features between the distress types, there may be other explanations for the low prediction scores across most of the classes. These include the inclusion of bounding boxes for ultra-thin hairline cracks and dataset imbalance. Although an existing issue in deep learning development, these cracks have a width of just a few pixels and are hard for the model to detect within a 4096 × 2048-pixel image, as shown in Figure 8, where a block crack in the center goes undetected for the image to the left [8]. The resizing of the images to a lower resolution also increases the impact of these missed detections as the features of the hairline cracks are further dissolved into the background pavement surface. Dataset imbalance is visible in the mAP50 scores as transverse cracks yielded a score of 0.523, and joints had relatively higher scores than the other distresses. The lack of a proper number of instances in both sealed cracks leads to their low detection scores. However, major distresses, such as alligator cracks and failures, are able to receive similar results without as many instances in the current dataset due to their easily visible features.

Another issue facing the detection accuracy of YOLOv5 with this dataset is the distress density within multiple images. Images of the pavement surface may be populated with multiple distresses and non-distresses, often creating input data with 10 or more bounding boxes. The high number of ground-truths in a small area of an image can cause confusion for the proposed YOLOv5 in effectively making correct decisions in predicting and classifying the distresses, along with the same issues in hairline cracks when the resolution is lowered. Along with changes to improve the current dataset, updating the computational hardware to develop the model with full-size images and combining features of both 2D and 3D images are being considered as measures to improve the model performance. More scenarios using different data augmentation skills will be tested to determine the most effective technique for model training. Also, other activation functions will be studied in our future work.

These initial results are still premature as this YOLOv5 model is not accustomed to the distress density and severity of coastal pavements. Efforts are being made to improve the development of the model by optimizing the ground-truths in the dataset for simpler, adequate detections to be made. Expansion of the dataset using more sections of the collected data with a focus on implementing more uncommon distresses, including failures, block and alligator cracks, and sealed cracking, should reduce the issue of dataset imbalance between the classification labels given to the proposed YOLOv5. Along with changes to improve the current dataset, updating the computational hardware to develop the model with full-size images and combining features of both 2D and 3D images are being considered as measures to improve the model’s accuracy.

## 5. Conclusions

YOLOv5 is a state-of-the-art model that is suitable for the task of pavement surface distress detection and classification. We developed the YOLOv5 models with a customized 2D/3D dataset for flexible pavements. The adequacy of this dataset and sophisticated development techniques are evaluated in this study. Although the mAP50 scores with the current dataset are lower than expected, there are several conclusions for this research:
The proposed YOLOv5 models are consistent with distress detection across most of the training scenarios, scoring 0.4–0.5 with most classes. The models perform the highest in the detection of joints and the lowest in the block cracks. The discrepancy between the two distress types is because of the number of instances for each class in the current dataset and how their defined features are insufficient. Joints are wider and more linear, and block cracks lack the major characteristics of alligator cracks and failures.The performance to detect sealed transverse and sealed longitudinal cracks is not high due to their low number of instances in the current dataset. The direction and implementation of cracking sealants make it hard for the proposed YOLOv5 to determine the type of sealed cracks.The detection of thin hairline cracks is a big challenge to the model since there are multiple instances for those wide transverse and longitudinal cracks within the dataset. Also, hairline cracks are presented in highly dense regions of distress that are difficult for the YOLOv5 to adequately process.Image resizing during model development reduces the resolution of the pavement surface, making it more difficult for the model to identify distresses that are shown on an image with a few pixels.

Size and representation of the training dataset, image quality, and computational resources may have led to lower scores for some classes. Also, this is a unique task for the models due to the lack of major differences in the appearance of the distresses. To resolve the issues, the proposed YOLOv5 could perform better in pavement surface distress detection and classification in the following ways:
Revisit the dataset and improve the annotations to be as consistent and adequate as possible for easier interpretation in the YOLOv5, as performed by Gong et al. [15].Expand the current dataset to include more images from pavement sections collected with the survey vehicle in Figure 1.Implement more instances of uncommon distresses such as block cracking, alligator cracks, sealed cracking, and failures.Combine features of 2D images with 3D images for complete visualization of road surface in an effort to improve model accuracy.Update the computational hardware to use original image input dimensions to retain the 2D/3D resolution from data collection.Use learning models with different network architectures in tests for the search for the best model for pavement distress detection and evaluation.

## Figures and Tables

**Figure 1 sensors-25-01145-f001:**
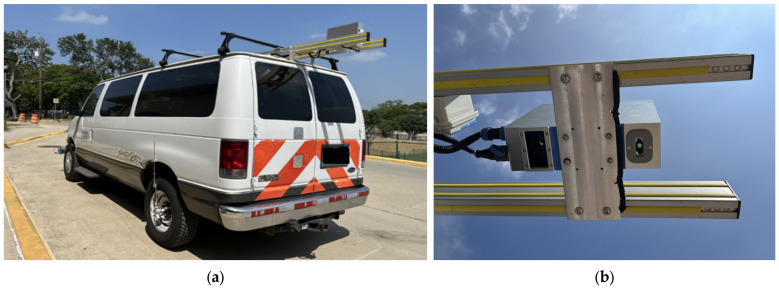
The 2D/3D pavement data collection equipment: (**a**) is the pavement survey vehicle with laser sensor attached from rear roof; (**b**) is the HyMIT laser sensor with line laser emitter and camera.

**Figure 2 sensors-25-01145-f002:**
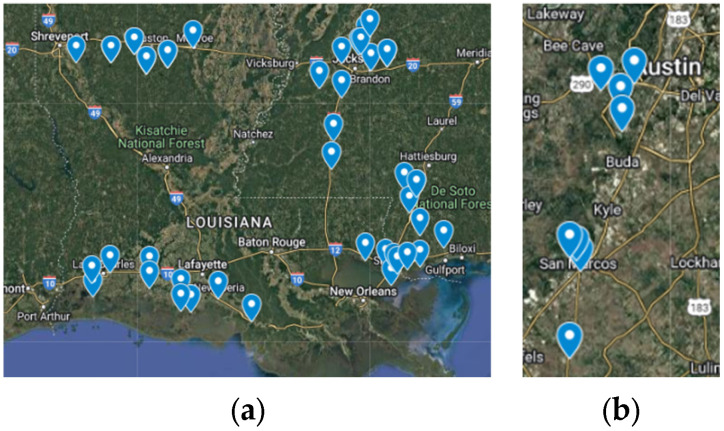
Location of pavement surface data collection sites in coastal states in United States of America: (**a**) is the pavement sections in Louisiana and Mississippi; (**b**) is the pavement sections in Texas.

**Figure 3 sensors-25-01145-f003:**
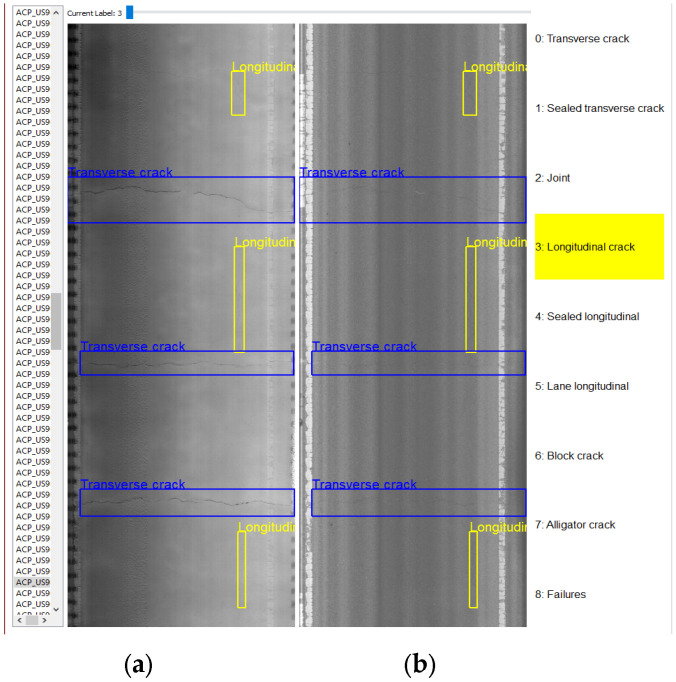
Custom annotation software interface: (**a**) is a 3D range image of pavement surface; (**b**) is a 2D intensity image of pavement surface. Both images have identical bounding box annotations, with pixel brightness in (**a**) indicating depth into or from pavement surface.

**Figure 4 sensors-25-01145-f004:**
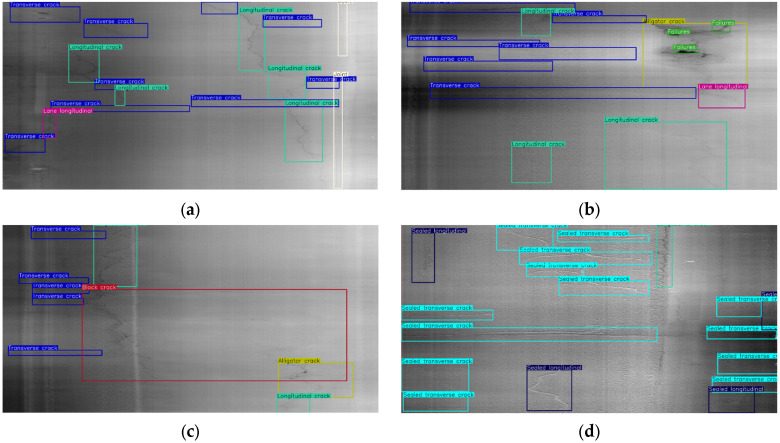
Labeled distresses and common objects in 3D images from the dataset: (**a**) is an asphalt surface with multiple transverse cracks, longitudinal cracks, lane longitudinal cracks, and joints; (**b**) is the presence of surface failures and alligator cracks with thin hairline cracks; (**c**) is a large block crack covering majority of lane width; (**d**) is sealed transverse and sealed longitudinal crack appearance on 3D images.

**Figure 5 sensors-25-01145-f005:**
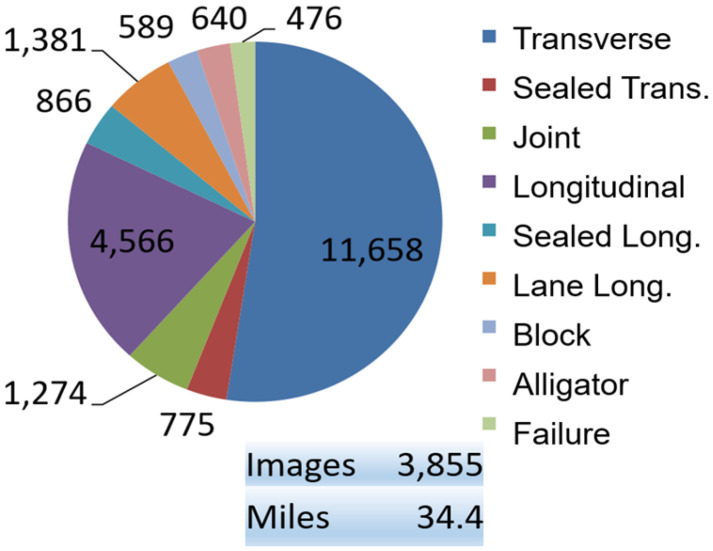
Distribution of labels in current dataset.

**Figure 6 sensors-25-01145-f006:**
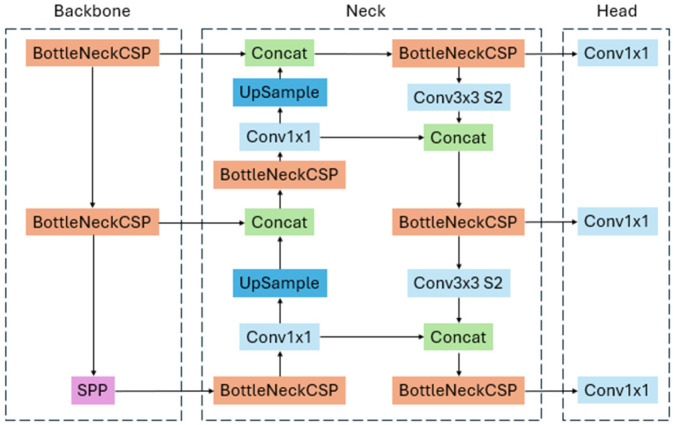
YOLOv5 architecture with cross-stage partial networks (BottleNeckCSP), spatial pyramid pooling (SPP), concatenation (Concat), convolution (Conv1x1, Conv3x3), and upsampling (UpSample).

**Figure 7 sensors-25-01145-f007:**
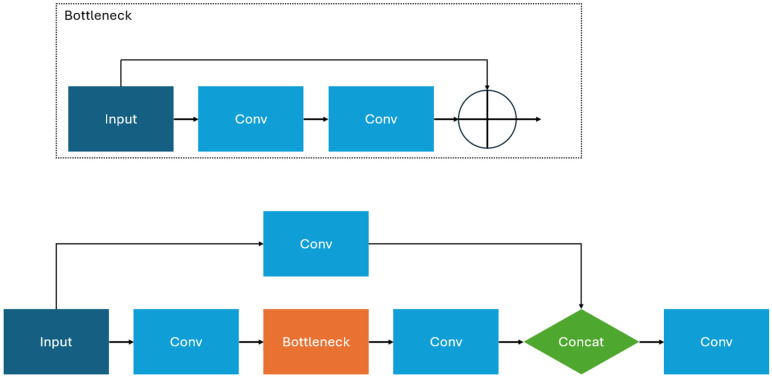
BottleNeckCSP architecture. Inputs in both BottleNeckCSP and Bottleneck module are split into two parts. One part goes through a convolutional layer, while the other is sent through multiple before rejoining in the Concatenation module.

**Figure 8 sensors-25-01145-f008:**
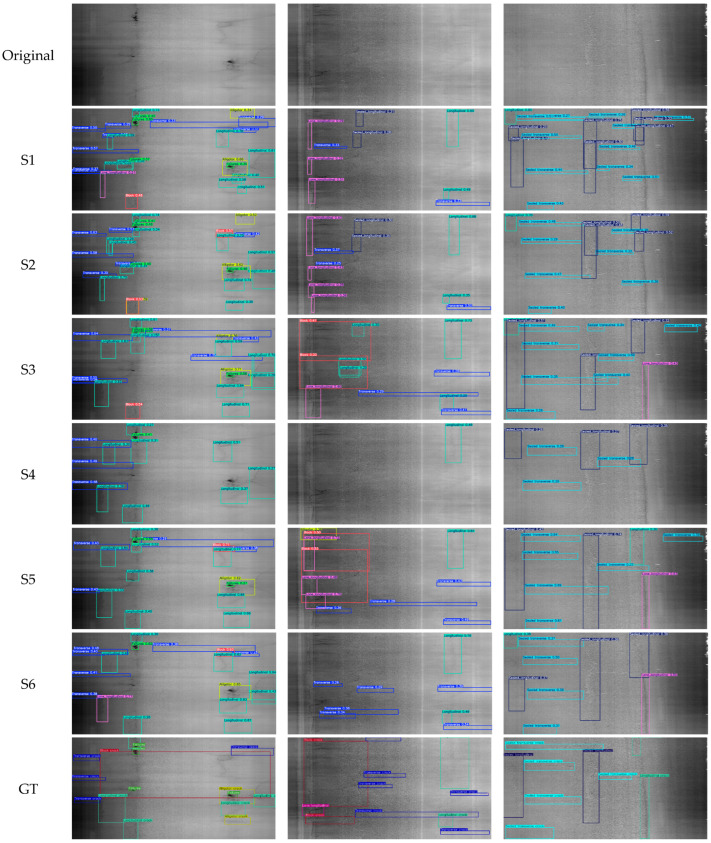
Prediction boxes generated via YOLOv5 detection. Original surface images, predictions from Scenarios 1–6 (S1 to S6), and the ground truth (GT) are displayed with color-coded boxes: cracks of transverse (blue), sealed transverse (cyan), longitudinal (aquamarine), sealed longitudinal (dark blue), lane longitudinal (pink), block (red), alligator (yellow), and failures (green). Joints (white) are not visualized.

**Table 1 sensors-25-01145-t001:** Classification list for pavement surface image dataset.

Distress Items	Non-Distress Items
Transverse Cracks Sealed Transverse Cracks Longitudinal Cracks Sealed Longitudinal Cracks Block Cracks Alligator Cracks	Joints Lane Longitudinal Cracks
Failures	

**Table 2 sensors-25-01145-t002:** Parameters for training scenarios of YOLOv5 with current dataset.

Training Scenario	Image Size	Batch Size	Augmentation
1	640 × 640	4	None
2	640 × 640	32	None
3	1024 × 1024	16	None
4	1024 × 1024	8	All augmentations activated
5	1024 × 1024	8	Flip probability = 0.5
6	1024 × 1024	8	Flip probability = 0.75

**Table 3 sensors-25-01145-t003:** Default data augmentation values in hyperparameter script used in training scenarios.

Augmentation	Description	Default Value	Unit
hsv_h	Alters the image’s hue	0.015	Fraction
hsv_s	Alters the image’s saturation	0.7	Fraction
hsv_v	Alters the image’s brightness value	0.4	Fraction
degrees	Rotates the image	0.0	Degrees
translate	Translates the image	0.1	Fraction
scale	Changes image scale	0.5	Gain
shear	Shifts rows and columns of image	0.0	Degrees
perspective	Changes original image perspective	0.0	Fraction (range of 0–0.001)
flipud	Vertical flip probability of image	0.0	Probability
fliplr	Horizontal flip probability of image	0.5	Probability
mosaic	Resizes and splits four images into one	1.0	Probability

**Table 4 sensors-25-01145-t004:** mAP50 scores of each training scenario.

Class Label	1	2	3	4	5	6
Transverse Cracks	0.516	0.497	0.507	0.228	**0.523**	0.507
Sealed Transverse Cracks	0.185	0.237	**0.473**	0.103	0.461	0.396
Joints	0.791	0.754	0.818	0.31	**0.838**	0.807
Longitudinal Cracks	0.366	0.386	**0.436**	0.327	0.409	0.413
Sealed Longitudinal Cracks	0.231	0.257	0.282	0.192	**0.325**	0.261
Lane Longitudinal Cracks	0.372	0.378	0.407	0.314	**0.474**	0.473
Block Cracks	0.066	0.058	0.168	0.0209	0.143	**0.177**
Alligator Cracks	0.406	**0.436**	0.381	0.288	0.42	0.423
Failures	0.445	0.484	0.52	0.335	**0.562**	0.48
General Score	0.375	0.387	0.444	0.226	**0.462**	0.437

Bold text indicates class label has the highest mAP50 score out of any scenario.

## Data Availability

The datasets presented in this article are not readily available because the data is part of an ongoing study or due to technical limitations, but we will let researchers know if the dataset is published in the future.

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
