# Peer review of "Detection of Flexible Pavement Surface Cracks in Coastal Regions Using Deep Learning and 2D/3D Images"

_sensors, 2025, doi:10.3390/s25041145_

Round 1

Reviewer 1 Report

Comments and Suggestions for Authors

The paper focuses on detecting pavement surface cracks in coastal regions using the YOLOv5 deep learning model trained on 2D/3D image datasets. It highlights the challenges of automated pavement distress detection, such as dataset imbalance, low-resolution images, and the presence of background noise, which impact model accuracy. The study evaluates the performance of YOLOv5 under various augmentation scenarios and identifies its limitations in detecting certain crack types, such as sealed or block cracks. In general, the motivation for this submission is easy to understand, but the novelty is limited. In addition, there are still several major weaknesses, as follows:

1. While YOLOv5 is a capable model, the paper does not justify its selection over newer or alternative models, such as YOLOv8, DETR, or Transformer-based models. The reasoning provided (extensive research on YOLOv5) is insufficient in the context of rapid advancements in object detection.

2. The dataset is heavily skewed towards transverse and longitudinal cracks, with fewer samples for block cracks, alligator cracks, and failures. This imbalance likely biases the model towards common classes while underperforming on minority classes. In addition, although YOLOv5 is pretrained on the COCO dataset, the methodology does not discuss how well this pretraining aligns with the domain-specific features of pavement crack detection.

3. The analysis of results lacks depth. For instance, the poor performance on block cracks is mentioned without exploring whether it is due to insufficient features, annotation errors, or dataset imbalance.

4. Some sections, such as the description of YOLOv5, are overly detailed and repetitive, which detracts from the readability of the paper. Key results presented in tables and figures (e.g., Table 4, Figure 6) are inadequately explained. Their connection to the conclusions is not sufficiently discussed. In addition, the conclusions merely restate the results without reflecting on the study's limitations or providing insights into its broader implications.

5. Some works about YOLO and object detection should be cited in this paper to make this submission more comprehensive, such as 10.1109/TIP.2023.3297408, 10.1109/TPAMI.2024.3511621, 10.1109/TII.2024.3352232.

Author Response

Reviewer #1

The paper focuses on detecting pavement surface cracks in coastal regions using the YOLOv5 deep learning model trained on 2D/3D image datasets. It highlights the challenges of automated pavement distress detection, such as dataset imbalance, low-resolution images, and the presence of background noise, which impact model accuracy. The study evaluates the performance of YOLOv5 under various augmentation scenarios and identifies its limitations in detecting certain crack types, such as sealed or block cracks. In general, the motivation for this submission is easy to understand, but the novelty is limited. In addition, there are still several major weaknesses, as follows:

Response: Thank you very much for sharing your insights.  Yes, this research has its merits but there are some limitations in it. We hope our clarifications and the revisions have improved the paper significantly so that readers can benefit from our work.

Comments 1: While YOLOv5 is a capable model, the paper does not justify its selection over newer or alternative models, such as YOLOv8, DETR, or Transformer-based models. The reasoning provided (extensive research on YOLOv5) is insufficient in the context of rapid advancements in object detection.

Response 1: Thank you. In the previous research work of [4] and [15], especially our research team already showed that YOLOv5 was the best model to perform distress detection, in which lower-definition 2D/3D images were curated in our datasets. We will continue to test the latest machine learning models along with the YOLOv5 model in our ongoing study. We made it clearer for why we chose YOLOv5 in Lines 220-226 on Page 6 in the manuscript.

“YOLOv5 has extensive community support stemming from years of development along with its ease of use due to its maturity and hardware compatibility. In addition, our previous research study in a similar scenario [15] showed YOLOv5 was the best performing model for our dataset and the computational environment setting. It is in our plan to apply the YOLOv5 and other models on the image data of different resolution levels using different data augmentation techniques to decide the best strategy for our next study, especially with limited computational resources.”

Comments 2: The dataset is heavily skewed towards transverse and longitudinal cracks, with fewer samples for block cracks, alligator cracks, and failures. This imbalance likely biases the model towards common classes while underperforming on minority classes. In addition, although YOLOv5 is pretrained on the COCO dataset, the methodology does not discuss how well this pretraining aligns with the domain-specific features of pavement crack detection.

Response 2: Thank you. Yes, the imbalance of the dataset is evident, and the presence of numerous transverse and longitudinal cracks reflect how often they appear along highway surfaces. During the dataset selection process, an effort was made to increase the instances of uncommon distresses such as alligator cracks, block cracks, and failures. But most of the images that contain the rare distress types also include numerous instances of transverse and longitudinal cracks. We have, accordingly, modified the methodology in Lines 261-267 on Page 8 to clarify how the pre-training on the COCO dataset can align with pavement distress detection.

“The COCO dataset contains 80 common object classes but do not contain labels or tasks for pavement distress detection. However, similar studies like what Gong et al. did in [15] show that YOLO models can have effective transfer-learning between the COCO dataset and pavement distress detection. This study takes the same strategy. In addition, data augmentation techniques are introduced into this research to increase the number of training data, reduce the imbalance problem among the annotated data, and improve the robustness of our models.”

Comments 3: The analysis of results lacks depth. For instance, the poor performance on block cracks is mentioned without exploring whether it is due to insufficient features, annotation errors, or dataset imbalance.

Response 3: We revised the initial results and discussion parts in Lines 350 through 365 on Page 11. This includes moving the second sentence to before the discussion on alligator cracks and failures.

“As for block cracks, there are fewer instances than other distresses due to the imbalance of the dataset on longitudinal cracks, transverse cracks, and the rest of the classes. Block cracks also lack key defining features from alligator cracks and failures, such as large areas of crack webs or deep spots. Meanwhile, alligator cracks and failures could be similar to each other on some annotated images. Failures, despite having unique characteristics in the form of potholes, have multiple appearances that vary based on the distress width. The limited number of failures in the current dataset present a wide variety of appearances such as features being slightly widened cracks, leading to their highest mAP50 scores being 0.562. These variations caused confusion for the proposed YOLOv5 models to effectively detect the failures as deep and eroded surface regions. Detection of longitudinal and lane longitudinal cracks performs similar to each other due to sharing similar features as opposed to joints, with scores of 0.436 and 0.476, respectively. Because their erratic patterns in each direction on the pavement surface make it difficult to detect properly, both sealed crack types are poorly detected with the trained YOLOv5 models, yielding scores of 0.473 for sealed transverse cracks and 0.325 for sealed longitudinal cracks.“

Comments 4: Some sections, such as the description of YOLOv5, are overly detailed and repetitive, which detracts from the readability of the paper. Key results presented in tables and figures (e.g., Table 4, Figure 6) are inadequately explained. Their connection to the conclusions is not sufficiently discussed. In addition, the conclusions merely restate the results without reflecting on the study's limitations or providing insights into its broader implications.

Response 4: Agreed. The description of YOLOv5 in the methodology has been reduced to skip details into the convolution or activation processes of YOLOv5, as those aspects were not modified in our study. However, we believe that the introduction given to the YOLOv5 model is important as it explains how our input data is evaluated throughout the model’s architecture. We have also made changes in the initial results and the discussion to further explain the results presented in the study.

The section explaining the convolution and activation functions was revised on Page 8, Lines 267 to 269:

“Activation functions such as sigmoid linear unit (SiLU) are used to introduce nonlinearity during convolution while sigmoid activation is introduced at the end of feature extraction for object class probability [22, 23].”

On Page 11, Lines 350 to 365 were revised as follows:

“As for block cracks, there are fewer instances than other distresses due to the imbalance of the dataset on longitudinal cracks, transverse cracks, and the rest of the classes. Block cracks also lack key defining features from alligator cracks and failures, such as large areas of cracks webs or deep spots. Meanwhile, alligator cracks and failures could be similar to each other on some annotated images. Failures, despite having unique characteristics in the form of potholes, have multiple appearances that vary based on the distress width. The limited number of failures in the current dataset present a wide variety of appearances such as features being slightly widened cracks, leading to their highest mAP50 scores being 0.562. These variations caused confusion for the proposed YOLOv5 models to effectively detect the failures as deep and eroded surface regions. Detection of longitudinal and lane longitudinal cracks performs similar to each other due to sharing similar features as opposed to joints, with scores of 0.436 and 0.476, respectively. Because their erratic patterns in each direction on the pavement surface make it difficult to detect properly, both sealed crack types are poorly detected with the trained YOLOv5 models, yielding scores of 0.473 for sealed transverse cracks and 0.325 for sealed longitudinal cracks.”

Similarly, revision was made in Lines 368-376:

“These are the inclusion of bounding boxes for ultra-thin hairline cracks and dataset imbalance. Although an existing issue in deep learning development, these cracks have a width of just a few pixels and are hard for the model to detect them within a 4,096 x 2,048-pixel image, shown in Figure 6 where a block crack in the center goes undetected for the image to the left [8]. The resizing of the images to a lower resolution also increases the impact of these missed detections as the features of the hairline cracks are further dissolved into the background pavement surface. Dataset imbalance is visible in the mAP50 scores as transverse cracks yielded a score of 0.523, and joints had relatively higher scores than the other distresses.”

On Page 13, Lines 431 to 435 were revised:

“Size and representatives of the training dataset, image quality, and computational resources may have led to the lower scores for some classes. Also, this is a unique task for the models due to lack of major differences in the appearance of the distresses. To resolve the issues, the proposed YOLOv5 could perform better on pavement surface distress detection and classification through the following ways:”

Comments 5: Some works about YOLO and object detection should be cited in this paper to make this submission more comprehensive, such as 10.1109/TIP.2023.3297408, 10.1109/TPAMI.2024.3511621, 10.1109/TII.2024.3352232

Response 5: Thank you for providing this information. All the references provide great insights into the development of machine learning using image data. The introduction part of the paper was revised to incorporate these references that explain more about YOLO and object detection.

The contents on Page 2, Lines 66 to 74 were revised as follows:

“Subtle distinctions of features between objects provide difficulty for AI models, but they can

be circumvented with proper techniques. Qi et al. utilized a novel visual blurriness network to properly detect glass surfaces from image data based on its intrinsic property to slightly distort light [4]. Along with this, visible light has its limitations when visualizing these subtle features. Other frequencies of visible data can provide much needed context for AI models, as Tang et al. had studied. Their research incorporated modalities that processed visual RBG and thermal data separately before integrating both for feature extraction, successfully improving detection performance of their model [5].”

On Page 3, Lines 114 to 119 were revised as follows:

“Quan et al. developed a Central Feature Pyramid (CFP) network to focus on features in corner regions of input images to improve predictions on the MS COCO 2017 dataset. Modifications to YOLOv5 and YOLOX were made using this CFP network, with an objective for utilizing deep features to regulate shallow, central features. This implementation of the CFP network provides enhanced predictions in the models [16].”

Reviewer 2 Report

Comments and Suggestions for Authors

(1)

According to page 4 of the study, a total of 40,000 images were collected after covering 408 miles in Louisiana and Mississippi, along with an additional 3,392 images from Texas. However, only 3,855 images were ultimately utilized for training, which accounts for approximately 11% of the total dataset. Could you elaborate on the criteria used to select this subset? Specifically, was the selection based on factors such as image quality, data relevance, computational efficiency, or preprocessing constraints?

(2)

Table 1 categorizes various types of cracks, but the distinction between these categories is not entirely clear without additional context. Given that the model’s performance depends on accurate classification of distress types, it would be beneficial to include representative images or schematic diagrams illustrating each crack type. Would it be possible to provide such visual references to enhance clarity? Doing so would help ensure that the classification scheme is well-defined and consistently interpreted.

(3)

Figure 5 presents the workflow of YOLOv5, highlighting the BottleNeckCSP module, which is crucial for computational efficiency and feature extraction. To enhance clarity, would it be possible to provide a detailed explanatory figure illustrating the structure and function of the BottleNeckCSP module within the YOLOv5 architecture?

(4)

The study mentions the use of two activation functions but does not specify whether alternative options were considered. Given their impact on training dynamics, convergence speed, and overall performance, were other activation functions, such as Mish, ReLU, or ELU, tested?

(5)

According to Table 4, the mAP50 score is highest in Training Scenario 5, highlighting the impact of data augmentation on model accuracy. However, the study does not provide a visual comparison between original images and their augmented counterparts. Would it be possible to include such a figure to illustrate how the dataset was modified?

(6)

While the improvement in Scenario 5 is evident, has the study quantified the individual contributions of different augmentation techniques? If multiple strategies were applied simultaneously, was an ablation study conducted to isolate their effects? Clarifying this would help validate the specific role of each augmentation method in enhancing model performance.

(7)

YOLO is widely recognized as a 1-stage detector, offering fast inference speed but generally lower accuracy compared to 2-stage detectors. Given this trade-off, are there any future plans to enhance accuracy while maintaining efficiency?

(8)

Additionally, in its current implementation, how long does the model take to perform classification and object detection in real-world scenarios?

Author Response

Reviewer #2

Comments 1: According to page 4 of the study, a total of 40,000 images were collected after covering 408 miles in Louisiana and Mississippi, along with an additional 3,392 images from Texas. However, only 3,855 images were ultimately utilized for training, which accounts for approximately 11% of the total dataset. Could you elaborate on the criteria used to select this subset? Specifically, was the selection based on factors such as image quality, data relevance, computational efficiency, or preprocessing constraints?

Response 1: Thank you for your comment. The reason why there is currently a subset that contains around 11% of the total dataset used for the study was to create a representative but yet more balanced data library for the YOLOv5 model training. During the annotation process, it was noticed that there was an overwhelming amount of transverse and longitudinal cracks within the dataset. As more images were labelled, focus was shifted to images that contain uncommon distresses such as alligator cracks, block cracks, and failures. Along with distress class selection, image quality was also considered. It led to the exclusion of some images, including those with heavy noise, bright/dark spots, poor lighting, too abundant distresses, and newly constructed pavements. We have revised the methodology part explaining the selection criteria in Lines 216-217 on Page 6:

“This subset of the full dataset was selected based on image quality and acquiring a substantial amount of instances for each distress class.”

Comments 2: Table 1 categorizes various types of cracks, but the distinction between these categories is not entirely clear without additional context. Given that the model’s performance depends on accurate classification of distress types, it would be beneficial to include representative images or schematic diagrams illustrating each crack type. Would it be possible to provide such visual references to enhance clarity? Doing so would help ensure that the classification scheme is well-defined and consistently interpreted.

Response 2: Thank you for the suggestion. It is essential to this study that the distresses and common objects are visualized rather than the text description or being listed in Table 1. We agree that by providing representative images from the dataset, the reader can better understand how each distress is labeled. Along with this, the visualization of the surface data can establish some of the challenges faced when establishing the dataset such as labeling based on crack orientation and the appearance of thin hairline cracks. An additional figure (Figure 4 on Page 6) was added showing the visualization of these objects in the image data.

The Lines 193 to 197 on Page 5 was revised as follows:

“The class list in Table 1 is used to label the distresses and common objects visible in the

images, as shown in Figure 4. The distresses can appear as singular cracks either parallel or perpendicular to the road’s centerline. They can also appear as an area of sparsely or tightly connected cracks such as block cracks or alligator cracks, respectively, or deep, dark regions on the surface in the dataset.”

The caption for the figure is shown in Lines 210-214 on Page 6:

              “Figure 4. Labeled distresses and common objects in 3D images from the dataset: (a)Asphalt surface with multiple transverse cracks, longitudinal cracks, lane longitudinal cracks and joints; (b) Presence of surface failures and alligator cracks with thin hairline cracks; (c) Large block crack covering majority of lane width; (d) Sealed transverse and sealed longitudinal cracks appearance on 3D images. “

Comments 3: Figure 5 presents the workflow of YOLOv5, highlighting the BottleNeckCSP module, which is crucial for computational efficiency and feature extraction. To enhance clarity, would it be possible to provide a detailed explanatory figure illustrating the structure and function of the BottleNeckCSP module within the YOLOv5 architecture?

Response 3: Agreed. The process for the BottleNeckCSP is a unique feature of how the input data is processed in YOLOv5. It is essential to the model due to its function to go through the images in a single pass, which helps in the balance between speed and accuracy. We revised Section 2.2 of Model Selection in Lines 249 to 257 on Page 8, and added a caption to the figure of BottleNeckCSP’s architecture:

“Figure 7. BottleNeckCSP architecture. Inputs in both BottleNeckCSP and Bottleneck module are split into two parts. One part goes through a convolutional layer while the other is sent through multiple layers before rejoining in the Concatenation module.”

“In the backbone, BottleNeckCSP is used to split the input image into two parts. One part of the input is sent through convolution while the other part goes through convolution, bottleneck, and concatenation layers. This process, visualized in Figure 7, reduces the computational load by allocating most of the feature extraction to one part rather than the whole image. “

Comments 4: The study mentions the use of two activation functions but does not specify whether alternative options were considered. Given their impact on training dynamics, convergence speed, and overall performance, were other activation functions, such as Mish, ReLU, or ELU, tested?

Response 4: Thank you for the reminder. Yes, SiLU and Sigmoid are the activation function used in this study. We will explore different activation functions in our future study as shown in Lines 392 to 397 on Page 13:

“Along with changes to improve the current dataset, updating the computational hardware to develop the model with full-size images and combining features of both 2D and 3D images are being considered as measures to improve the model performance. More scenarios using different data augmentation skills will be tested to determine the most effective technique for the model training. Also, other activation functions will be studied in our future work.”

Comments 5: According to Table 4, the mAP50 score is highest in Training Scenario 5, highlighting the impact of data augmentation on model accuracy. However, the study does not provide a visual comparison between original images and their augmented counterparts. Would it be possible to include such a figure to illustrate how the dataset was modified?

Response 5: Thank you for mentioning this lack of comparison between the original and augmented images in this study. Yes, we provided a new figure (Figure 6 on Page 12) and explained it in Lines 368 to 376 on Page 11:

“These are the inclusion of bounding boxes for ultra-thin hairline cracks and dataset imbalance. Although an existing issue in deep learning development, these cracks have a width of just a few pixels and are hard for the model to detect them within a 4,096 x 2,048-pixel image, shown in Figure 6 where a block crack in the center goes undetected for the image to the left [8]. The resizing of the images to a lower resolution also increases the impact of these missed detections as the features of the hairline cracks are further dissolved into the background pavement surface. Dataset imbalance is visible in the mAP50 scores as transverse cracks yielded a score of 0.523, and joints had relatively higher scores than the other distresses.”

Comments 6: While the improvement in Scenario 5 is evident, has the study quantified the individual contributions of different augmentation techniques? If multiple strategies were applied simultaneously, was an ablation study conducted to isolate their effects? Clarifying this would help validate the specific role of each augmentation method in enhancing model performance.

Response 6: Thank you for the comment. Yes, Scenario 5 has a better performance over all the distresses when vertical and horizontal flips were implemented. In contrast, the augmentations in Scenario 4 include hue, saturation, and brightness values, geometry via scaling, translation, angularity, and perspective change on the input images, but the overall performance is the worst. This indicates that such augmentation techniques may not work for our collected data. Also, as stated in the Section of Data Collection, our data collection is along the driving direction for each lane, thus, the flipping of an image will create another training example. But it could cause lower accuracy if the flip probability increases from 0.5 to 0.75, or there are more flips on the labels (see Scenario 6). But we will test more scenarios to finalize the influences of individual data augmentation technique in our future study. A new sentence was added in Lines 392 to 397 on Page 13:

    “Along with changes to improve the current dataset, updating the computational hardware to develop the model with full-size images and combining features of both 2D and 3D images are being considered as measures to improve the model performance. More scenarios using different data augmentation skills will be tested to determine the most effective technique for the model training. Also, other activation functions will be studied in our future work.”

Comments 7: YOLO is widely recognized as a 1-stage detector, offering fast inference speed but generally lower accuracy compared to 2-stage detectors. Given this trade-off, are there any future plans to enhance accuracy while maintaining efficiency?

Response 7: Thank you for the observation. We do plan on improving the accuracy of the models using different network architectures. We added a new task to our future work list in Lines 446 to 447 on Page 14:

“Deep learning models with different network architectures will be tested for the search of the best model for pavement distress detection and evaluation.”

Comments 8: Additionally, in its current implementation, how long does the model take to perform classification and object detection in real-world scenarios?

Response 8: Due to the single-pass processing of the visual data in YOLOv5’s architecture, a fully trained model can make predictions for multiple images within a second. The inference speed of the YOLOv5 model is 34.3 ms per image. Lines 332 to 333 on Page 10 were revised as follows:

“Inference speed for the predictions took 34.3 ms per image.”

Round 2

Reviewer 1 Report

Comments and Suggestions for Authors

No more comments.

Reviewer 2 Report

Comments and Suggestions for Authors

The manuscript is properly revised based on the reviewer's comments and I suggest the publication of this paper in this journal.